# Analysis of Factors Influencing the Trophic State of Drinking Water Reservoirs in Taiwan

**Cheng-Wei Hung [1] and Lin-Han Chiang Hsieh [2,***

[1] Graduate Institute of Environmental Engineering, National Taiwan University, Taipei 106, Taiwan; hungjason111210@gamil.com
[2] Department of Environmental Engineering, College of Engineering, Chung Yuan Christian University, Zhongli 320, Taiwan
[*] Correspondence: chianghsieh@cycu.edu.tw; Tel.: +886-3-265-4908

**Abstract:** Eutrophication is an environmental pollution problem that occurs in natural water bodies. Regression analyses with interaction terms are carried out to identify the factors influencing the Shimen, Mingde, and Fongshan Reservoirs in Taiwan. The results indicate that the main factor influencing these reservoirs is total phosphorus. In the Shimen and Mingde Reservoirs, the influence of total phosphorus, when interacting with other factors, on water quality trophic state is more serious than that of total phosphorus per se. This implies that the actual influence of total phosphorus on the eutrophic condition could be underestimated. Furthermore, there was no deterministic causality between climate and water quality variables. In addition, time lagged effects, or the influence of their interaction with other variables, were considered separately in this study to further determine the actual relationships between water trophic state and influencing factors. The influencing patterns for three reservoirs are different, because the type, size, and background environment of each reservoir are different. This is as expected, since it is difficult to predict eutrophication in reservoirs with a universal index or equation. However, the multiple linear regression model used in this study could be a suitable quick-to-use, case-by-case model option for this problem.

**Keywords:** eutrophication; variable interactions; multiple linear regression; reservoir

## 1. Introduction

Constructing reservoirs is one of the most effective ways of storing water in Taiwan. Surface runoff may be caught during periods of high flow and provide water for people's livelihood, industry, and agriculture during periods of water shortage.

Eutrophication, the most challenging water pollution problem in water bodies, will eventually become an issue in many reservoirs [1]. Eutrophication negatively affects the water quality, safety, ecological integrity, and sustainability of global water resources [2–4]. It has long been believed that excessive phosphorus is the main reason for eutrophication [5]. However, population density, urbanization, and agricultural activities are also factors that influence the water quality of freshwater systems [6–10]. Since the 1940s, a substantial population increase, land-use intensification, and the use of agricultural fertilizers from developed countries [11], as well as the use of detergents containing phosphate compounds since the 1950s, have accelerated the eutrophication of waterbodies [12].

Eutrophication influences the water volume and quality in reservoirs. Regarding water volume, algae distributed on the water surface causes water hypoxia and a decrease in water transparency [13,14], leading to a substantial death of aquatic organisms, which are then deposited on the bottom of the reservoir, which, in turn, reduces the reservoir's capacity over time. Regarding water quality, the proliferation of algae causes algal blooms and releases algal poison, which both influences water quality conditions such as dissolved oxygen, transparency, odor, and pH value, and it also causes problems during the filtration of drinking water, increasing health risks to people [15,16]. Eutrophication also causes the

depletion of dissolved oxygen in water, which may have potentially harmful influences on the habitats of fish and macroinvertebrates [17,18]. In addition, eutrophication results in the release of nutrients such as phosphorus and ammonium into the water [19–21], potentially producing toxic heavy metal ions [22].

At present, reservoirs in Taiwan are facing a crisis of deteriorating water quality. According to the Environmental Water Quality Monitoring Annual Report of the Environmental Protection Agency, among the 26 reservoirs on the main island of Taiwan, 7 reservoirs are in a eutrophic state, 18 are in a mesotrophic state, and only 1 is in an oligotrophic state.

However, the Carlson trophic state index (CTSI) for assessing the water quality of reservoirs is not completely suitable in Taiwan. Taiwan is affected by frequent heavy rainfall, and particularly after typhoons and heavy rains, large amounts of sand, soil, and rock flow into the reservoirs, leading to a large increase in the concentration of suspended solids and a decrease in water transparency. Thus, it is quite likely that a water body with a high CTSI value, but without a large amount of algae distributed on the water surface, may be identified as eutrophic.

This study aims to investigate the correlation between weather and water quality factors and their degree of influence on the trophic state of reservoirs both as single variables and as interactions of variables. We also discuss the suitability of CTSI for assessing water quality in reservoirs in Taiwan. We used weather and water quality data from 2017 to 2019 from three main reservoirs in Taiwan: Shimen Reservoir, Mingde Reservoir, and Fongshan Reservoir. Chlorophyll a was used as an indicator to illustrate the degree of eutrophication, and data were analyzed using multiple linear regressions (MLR) including time lags and variable interactions.

## 2. Materials and Methods

### 2.1. Characteristics of Reservoir

The Shimen Reservoir is a stable source of water supply in northern Taiwan and is the third largest reservoir in Taiwan (Figure 1). It is a multi-objective water conservancy project that combines benefits such as irrigation, power generation, water supply, flood control, sightseeing, and recreation. The Shimen Reservoir has a catchment area of 763 km$^2$, a full water level of 8 km$^2$, a total storage capacity of 309 million m$^3$, and an effective storage capacity of 197 million m$^3$ [23]. The irrigation area of the Shimen Reservoir includes three counties: Hsinchu, Taoyuan, and New Taipei City. It provides for a daily consumption of 800,000 m$^3$ of livelihood water and also the Shimen Power Plant with 230 million kWh of power generation per year [24].

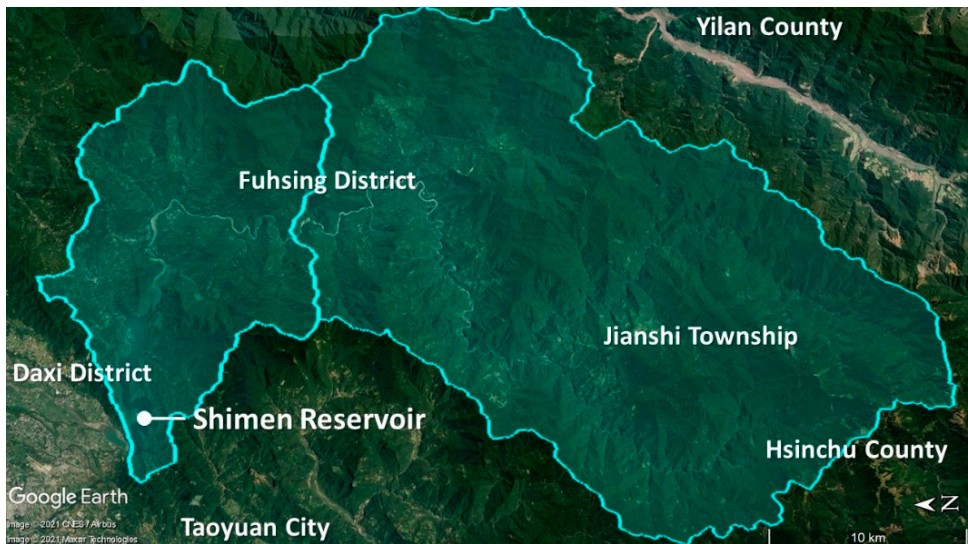

**Figure 1.** Location of Shimen Reservoir (including water catchment area).

The Mingde Reservoir provides more water for consumption in the Miaoli County, where there are more mountains and few fields (Figure 2). The Mingde Reservoir has a catchment area of 61 km$^2$, a full water level of 1.7 km$^2$, a total storage capacity of 17.7 million m$^3$, and an effective storage capacity of 12.2 million m$^3$. The irrigation area of the Mingde Reservoir is 13 km$^2$, and it provides 27,000 m$^3$ of water daily [25].

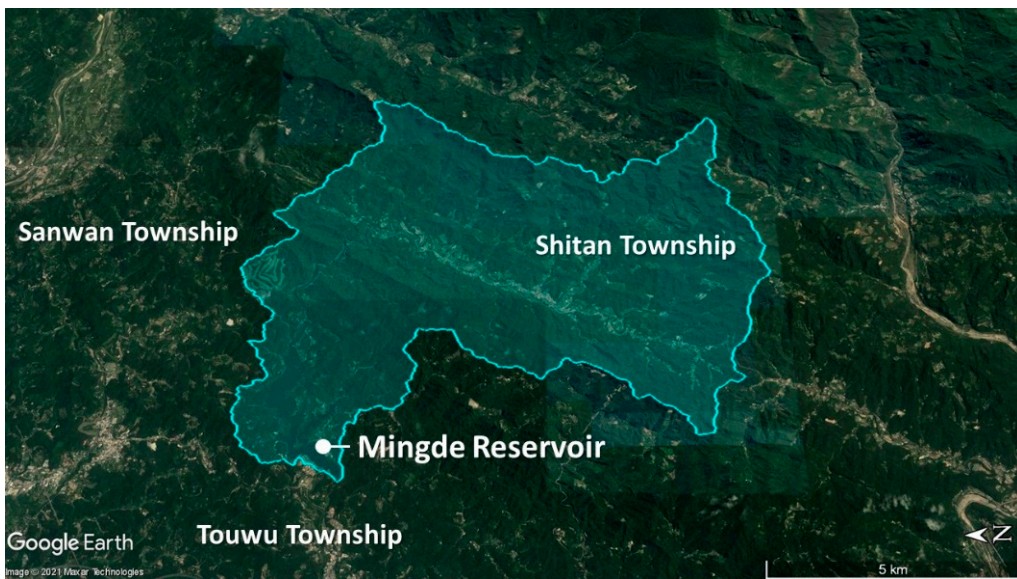

**Figure 2.** Location of Mingde Reservoir (including water catchment area).

The Fongshan Reservoir is an off-site reservoir located in Kaohsiung City and provides Kaohsiung with a large additional water supply because of the high population concentration and the rapid development of industry and commerce, which increases water consumption in the area (Figure 3). It has a catchment area of 2.75 km$^2$, a full water level of 0.75 km$^2$, a total storage capacity of 9.2 million m$^3$, and an effective storage capacity of 8.5 million m$^3$ [26]. The Fongshan Reservoir supplies 1.6 million tons of water daily, of which 350,000 tons, 22%, caters for industrial consumption [27].

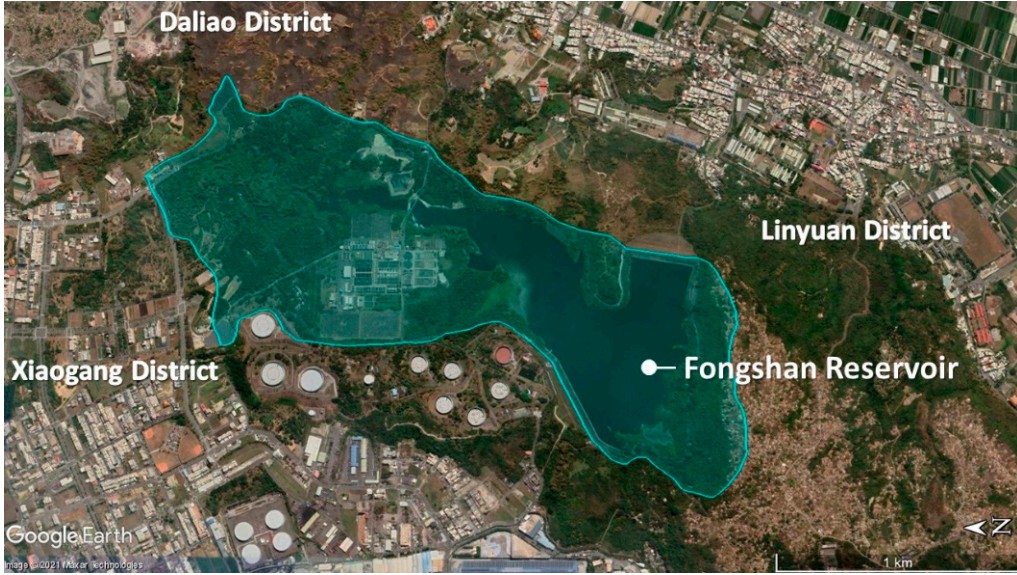

**Figure 3.** Location of Fongshan Reservoir (off-stream).

*2.2. Dataset*

The Taiwan Environmental Protection Administration (EPA) changed reservoir water quality monitoring from quarterly monitoring in previous years to monthly monitoring from January 2017. In this study, we used monthly weather and water quality data from the Shimen, Mingde, and Fongshan Reservoirs from January 2017 to December 2019.

Weather data included the daily statistics of rainfall (mm) and inflow (mm) data in the catchment area from 2017 to 2019 and was downloaded from the disaster prevention information service website of the Water Resources Agency (WRA). The data collection period corresponds to that of the EPA water quality monitoring data from each reservoir. In addition, monthly water temperature (WT) data from 2017 to 2019 were collected from the national water quality monitoring information website of the EPA [28].

Water quality data were collected from the monthly statistics on the national water quality monitoring information website of the EPA and included chlorophyll a (Chl-a), dissolved oxygen (DO), transparency (SD), total phosphorus (TP), pH, conductivity, suspended solids (SS), chemical oxygen demand (COD), and ammonia nitrogen (AN) sampled from 2017 to 2019 [28].

In order to investigate whether the different seasons affect the degree of influence, March to May were designated as Season 1 (S1), June to August were designated as Season 2 (S2), September to November were designated as Season 3 (S3), and December to February were designated as Season 4 (S4). S4 was used as a control, and S1 to S3 were analyzed to identify the degree of influence that each variable has on water quality in the different seasons.

*2.3. Methodology*

2.3.1. Regression Analysis

We used multiple linear regression (MLR) using regression analysis (Equation (1)). In order to identify how much each factor influences the eutrophication of the reservoirs, we used rapidly adjusting variables in regression models to analyze weather and water quality factors.

$$Y_i = \beta_0 + \beta_1 X_{1i} + \beta_2 X_{2i} + \cdots + \beta_k X_{ki} + \varepsilon_i \tag{1}$$

where $Y_i$ is the *i*-th observation value in the dependent variable, which represents the concentration of chlorophyll a in this study. The independent variables $X_{1i}$ to $X_{ki}$ are weather (rainfall, inflow, and water temperature) and water quality (Chl-a, DO, etc.) factors. $\beta_0$ is an intercept term, $\beta_1$ to $\beta_k$ are the slope terms, and also the unknown coefficients corresponding to the independent variables $X_{1i}$ to $X_{ki}$. $\varepsilon_i$ is a random error term.

2.3.2. Time-Lag

The reason of applying time-lag variables in this study is that the current weather or water quality factors do not necessarily have an immediate influence on the Chl-a concentration. These time lag situations might be one week, one month, or even more, so it is not suitable to use the monitoring current data in the regression analysis.

Basic analysis, Lag 1, and Lag 2 data in the regression model were selected using the following steps: Firstly, we select the significant data of the three types of data. Secondly, if more than one of the three types of data were significant, we selected the data collected closest to the monitoring date, i.e., basic analysis data take precedence over Lag 1 data, which takes precedence over Lag 2 data. Thirdly, if there was no significance in the three types of data, basic analysis data were selected as a representative term.

2.3.3. Interaction Terms

We also test whether the influence of weather and water quality factors on the water trophic state is affected by their interaction. The traditional ordinary least square (OLS) formula can be illustrated as shown in Equation (2):

$$Y = \beta_1 X_1 + \cdots + \beta_4 X_4 \tag{2}$$

Regarding the interaction of factors, we need to consider whether factors are correlated. We tested the pair-by-pair interaction of factors that are theoretically related using an MLR equation and then analyzed whether the correlation was still significant after the interaction. If it was significant, the two factors were grouped to form an interaction term that was added in the equation, and the correlation was analyzed as shown in Equations (3) and (4). $\beta_5 X_1 X_2$ and $\beta_6 X_3 X_4$ are the interaction terms added on the basis of an MLR.

$$Y_1 = \beta_1 X_1 + \cdots + \beta_4 X_4 + \beta_5 X_1 X_2 \tag{3}$$

$$Y_2 = \beta_1 X_1 + \cdots + \beta_4 X_4 + \beta_6 X_3 X_4 \tag{4}$$

If the individual analysis results of interaction terms $\beta_5 X_1 X_2$ and $\beta_6 X_3 X_4$ in the regressions of $Y_1$ and $Y_2$ were both significant, then the two factors were analyzed in the same regression formula for $Y_3$, as shown in Equation (5).

$$Y_3 = \beta_1 X_1 + \cdots + \beta_4 X_4 + \beta_5 X_1 X_2 + \beta_6 X_3 X_4 \tag{5}$$

If the regression results of $\beta_5 X_1 X_2$ and $\beta_6 X_3 X_4$ in $Y_3$ were both significantly correlated, the combination was retained, and the method was repeated on other interaction terms. At most, two interaction groups were included in the regression formula of each reservoir, and the factors in the two groups did not overlap with each other.

We used three conditions for selecting two groups of interaction terms with simultaneous significant correlations: Firstly, we considered TP and AN that represent the nutrient factors in the two interaction groups. Secondly, we considered WT, rainfall, and inflow that are representative of the weather factors, and WT had priority over rainfall, and rainfall had priority over inflow. Thirdly, we considered the $R^2$ value that could be explained by applying each group to the regression formula as a final selection step.

### 2.3.4. Interrelationship of Interaction Terms

Equation (2) was simplified by reducing variables and adding an interaction term, as shown in Equation (6).

$$Y = \beta_1 X_1 + \beta_2 X_2 + \beta_3 X_1 X_2 \tag{6}$$

Equation (6) was rewritten as Equation (7), in which other variables are fixed, $X_1$ increases by 1 unit, and $X_2$ does not increase, and the dependent variable becomes $Y_1$.

$$Y_1 = \beta_1 (X_1 + 1) + \beta_2 X_2 + \beta_3 (X_1 + 1) X_2 \tag{7}$$

Equation (7) minus Equation (6) provides Equation (8), which represents a situation in which other variables are fixed, $X_1$ increases by 1 unit and $X_2$ does not increase, and the unit amount $\Delta Y_1$ is the dependent variable that will increase.

$$\Delta Y_1 = Y_1 - Y = \beta_1 + \beta_3 X_2 \tag{8}$$

Similarly, if other variables are fixed but $X_1$ does not increase and $X_2$ increases by 1 unit, the dependent variable becomes $Y_2$ (Equation (9)). Equation (9) minus Equation (6) provides Equation (10), which represents the unit amount $\Delta Y_2$—the dependent variable that will increase.

$$Y_2 = \beta_2 + \beta_3 X_1 \tag{9}$$

$$\Delta Y_2 = Y_2 - Y = \beta_2 + \beta_3 X_1 \tag{10}$$

When both $X_1$ and $X_2$ increase by 1 unit, and other variables are fixed, Equation (11) is obtained. Equation (11) minus Equation (6) provides Equation (12), which represents the unit amount $\Delta Y_3$—the dependent variable that will increase.

$$Y_3 = \beta_1 (X_1 + 1) + \beta_2 (X_2 + 1) + \beta_3 (X_1 + 1)(X_2 + 1) \tag{11}$$

$$\Delta Y_3 = Y_3 - Y = \beta_1 + \beta_2 + \beta_3 + \beta_3(X_1 + X_2) \tag{12}$$

Equations (8), (10), and (12) were integrated (as shown in Table 1), where $\Delta X_1$ and $\Delta X_2$ are the unit amounts by which the independent variables, $X_1$ and $X_2$, are increased. Table 1 shows that when $X_1$ and $X_2$ are increased by 1 unit separately or simultaneously, the unit amount $\Delta Y$, which is the dependent variable, will increase.

**Table 1.** Interrelationship of interaction terms.

| $\Delta X_1$ | $\Delta X_2$ | $\Delta Y$ |
|:---:|:---:|:---:|
| 1 | 0 | $\beta_1 + \beta_3 X_2$ |
| 0 | 1 | $\beta_2 + \beta_3 X_1$ |
| 1 | 1 | $\beta_1 + \beta_2 + \beta_3 + \beta_3(X_1 + X_2)$ |

*2.4. Model*

In this study, Chl-a is set as the dependent variable, and the eleven weather and water quality factors, WT, DO, SD, TP, pH, conductivity, SS, COD, AN, rainfall, and inflow, are set as independent variables. STATA (version 13) is used in this study to perform correlation analysis and regression analysis.

The research model of this study is based on MLR. Using the result of a Hausman Test, we chose to include a random effects model in the MLR model to avoid or reduce ignoring differences between the data (which results in the omission of variables and leads to estimation errors) and also to reduce the occurrence of collinearity problems between variables.

The MLR model built in this study includes the random effects model and performs a time-lag analysis as well as adding specific interaction terms separately based on the difference in the reservoir data.

**3. Results and Discussion**

*3.1. Descriptive Statistics*

This study lists the descriptive statistics of the Shimen, Mingde, and Fongshan Reservoirs (Table 2). The minimums of TP, SS, COD, and AN were below the detection limit (ND).

**Table 2.** Descriptive statistics from the Shimen, Mingde, and Fongshan Reservoir data.

| Factors | | Chl-a | WT | DO | SD | TP | pH | Conductivity | SS | COD | AN | Rainfall | Inflow |
|:---:|:---:|:---:|:---:|:---:|:---:|:---:|:---:|:---:|:---:|:---:|:---:|:---:|:---:|
| Unit | | μg/L | °C | mg/L | m | mg/L | - | μmho/cm | mg/L | mg/L | mg/L | mm | cms |
| | SR | 4.078 | 23.741 | 8.906 | 1.909 | 0.021 | 8.396 | 210.282 | 3.808 | 3.818 | 0.021 | 5.642 | 302.692 |
| Mean | MR | 20.395 | 25.681 | 10.108 | 1.145 | 0.023 | 8.714 | 242.361 | 6.753 | 9.218 | 0.040 | 4.575 | 15.509 |
| | FR | 49.297 | 27.540 | 7.235 | 0.704 | 0.810 | 7.983 | 646.488 | 17.016 | 16.822 | 1.052 | 1.653 | 23.329 |
| | SR | 21.900 | 31.200 | 11.700 | 4.400 | 0.059 | 9.360 | 302.000 | 87.000 | 10.600 | 0.130 | 27.900 | 1114.350 |
| Max | MR | 54.400 | 32.700 | 16.500 | 2.000 | 0.033 | 9.720 | 368.000 | 28.000 | 15.400 | 0.120 | 101.100 | 150.320 |
| | FR | 258.000 | 32.100 | 14.500 | 1.300 | 1.710 | 8.780 | 1130.000 | 46.000 | 47.600 | 6.850 | 35.500 | 32.700 |
| | SR | 0.500 | 15.000 | 5.600 | 0.400 | ND | 7.000 | 150.000 | ND | ND | ND | 0.000 | 47.780 |
| Min | MR | 2.000 | 11.700 | 5.400 | 0.400 | 0.007 | 7.380 | 176.000 | 2.400 | ND | ND | 0.000 | 0.030 |
| | FR | 5.700 | 19.100 | 1.700 | 0.300 | 0.091 | 4.460 | 343.000 | 5.200 | 4.400 | ND | 0.000 | 9.800 |
| | SR | 2.878 | 4.413 | 1.085 | 0.768 | 0.011 | 0.584 | 26.936 | 6.346 | 3.421 | 0.019 | 7.226 | 260.253 |
| Std. Dev. | MR | 12.267 | 5.012 | 2.160 | 0.324 | 0.006 | 0.499 | 37.782 | 3.883 | 2.967 | 0.029 | 16.893 | 27.482 |
| | FR | 44.075 | 3.093 | 2.699 | 0.218 | 0.440 | 0.324 | 177.133 | 8.298 | 8.901 | 1.493 | 6.328 | 6.648 |

SR: Shimen Reservoir, MR: Mingde Reservoir, FR: Fongshan Reservoir, ND: The monitoring data is smaller than the detection limit

The average values of Chl-a, TP, Conductivity, SS, COD, and AN in the Fongshan Reservoir are several times more than the average values of the other two reservoirs. The concentration of Chl-a is 12 times that of the Shimen Reservoir and 2.4 times that of the Mingde Reservoir. The concentration of TP is 40 times that of the Shimen and Mingde Reservoirs. The concentration of AN is 52.5 times that of the Shimen Reservoir and 56.2 times that of the Mingde Reservoir.

The maximum Chl-a, TP, Conductivity, COD, and AN were recorded in the Fongshan Reservoir, the maximum SS was recorded in the Shimen Reservoir, and the maximum pH was recorded in the Mingde Reservoir. There were large variations in the daily rainfall and inflow data. It is possible that there was no rain on the day monitoring was conducted but heavy rainfalls the next day; therefore, we do not discuss the maximum and minimum rainfall and inflow.

The minimum DO of the Fongshan Reservoir was about three times lower than those of the Shimen and Mingde Reservoirs. However, its minimum TP concentration was still much larger than that of the Shimen and Mingde Reservoirs. The Fongshan Reservoir was the only reservoir with a pH of less than 7. The minimum conductivity of the Fongshan Reservoir was more than two times that of the Shimen and Mingde Reservoirs. The minimum SS of the Fongshan Reservoir was about two times that of the Mingde Reservoir.

### 3.2. Correlation Analysis

Results from the correlation analysis of the Shimen, Mingde, and Fongshan Reservoirs are show in Tables 3 and 4. The absolute value of the correlation coefficient was above 0.6, which can be regarded as a high correlation between the two factors.

**Table 3.** Correlation coefficients between factors from correlation analyses (1/2).

|  |  | Chl-a | WT | DO | SD | TP | pH | Conductivity | SS | COD | AN | Rainfall | Inflow |
|---|---|---|---|---|---|---|---|---|---|---|---|---|---|
| Chl-a | SR | 1.000 | | | | | | | | | | | |
| | MR | 1.000 | | | | | | | | | | | |
| | FR | 1.000 | | | | | | | | | | | |
| WT | SR | 0.266 | 1.000 | | | | | | | | | | |
| | MR | 0.208 | 1.000 | | | | | | | | | | |
| | FR | 0.241 | 1.000 | | | | | | | | | | |
| DO | SR | 0.260 | −0.126 | 1.000 | | | | | | | | | |
| | MR | −0.211 | 0.055 | 1.000 | | | | | | | | | |
| | FR | −0.022 | 0.355 | 1.000 | | | | | | | | | |
| SD | SR | 0.053 | −0.005 | 0.093 | 1.000 | | | | | | | | |
| | MR | −0.156 | 0.067 | 0.177 | 1.000 | | | | | | | | |
| | FR | −0.438 | −0.388 | −0.231 | 1.000 | | | | | | | | |
| TP | SR | 0.369 | −0.011 | 0.180 | −0.141 | 1.000 | | | | | | | |
| | MR | 0.019 | −0.034 | −0.180 | −0.126 | 1.000 | | | | | | | |
| | FR | 0.321 | −0.282 | −0.488 | −0.025 | 1.000 | | | | | | | |
| pH | SR | 0.308 | 0.700 | 0.297 | 0.079 | 0.114 | 1.000 | | | | | | |
| | MR | −0.011 | 0.591 | 0.634 | 0.145 | −0.008 | 1.000 | | | | | | |
| | FR | 0.042 | 0.549 | 0.700 | −0.212 | −0.549 | 1.000 | | | | | | |

**Table 4.** Correlation coefficients between factors from correlation analyses (2/2).

|  |  | Chl-a | WT | DO | SD | TP | pH | Conductivity | SS | COD | AN | Rainfall | Inflow |
|---|---|---|---|---|---|---|---|---|---|---|---|---|---|
| Conductivity | SR | −0.369 | −0.200 | −0.288 | −0.237 | 0.004 | −0.413 | 1.000 | | | | | |
| | MR | −0.191 | −0.644 | −0.296 | −0.233 | −0.049 | −0.649 | 1.000 | | | | | |
| | FR | 0.206 | −0.210 | −0.495 | 0.021 | 0.854 | −0.543 | 1.000 | | | | | |
| SS | SR | −0.029 | −0.118 | −0.105 | −0.350 | 0.317 | −0.160 | 0.267 | 1.000 | | | | |
| | MR | 0.108 | −0.154 | −0.137 | −0.575 | 0.088 | −0.250 | 0.372 | 1.000 | | | | |
| | FR | 0.352 | 0.261 | 0.101 | −0.600 | −0.051 | 0.146 | −0.138 | 1.000 | | | | |
| COD | SR | −0.039 | 0.315 | 0.075 | 0.019 | −0.108 | 0.427 | −0.076 | −0.029 | 1.000 | | | |
| | MR | 0.440 | 0.257 | −0.139 | −0.131 | −0.079 | 0.077 | −0.049 | 0.171 | 1.000 | | | |
| | FR | 0.602 | 0.089 | −0.067 | −0.488 | 0.599 | −0.030 | 0.454 | 0.473 | 1.000 | | | |
| AN | SR | 0.193 | 0.091 | 0.115 | −0.009 | 0.078 | 0.220 | −0.110 | −0.011 | 0.197 | 1.000 | | |
| | MR | 0.224 | 0.036 | −0.186 | −0.044 | 0.110 | −0.099 | 0.108 | 0.221 | 0.074 | 1.000 | | |
| | FR | 0.253 | −0.180 | −0.402 | −0.140 | 0.827 | −0.479 | 0.696 | −0.007 | 0.469 | 1.000 | | |
| Rainfall | SR | −0.078 | 0.121 | −0.455 | −0.037 | 0.107 | −0.003 | 0.082 | 0.171 | 0.054 | −0.101 | 1.000 | |
| | MR | −0.179 | 0.014 | 0.064 | 0.044 | 0.120 | 0.110 | 0.093 | 0.068 | 0.007 | −0.087 | 1.000 | |
| | FR | −0.045 | 0.071 | −0.233 | 0.160 | 0.091 | −0.231 | −0.051 | 0.049 | −0.005 | 0.026 | 1.000 | |
| Inflow | SR | 0.332 | 0.096 | 0.086 | 0.003 | 0.356 | 0.275 | −0.354 | 0.027 | 0.092 | 0.220 | 0.262 | 1.000 |
| | MR | 0.055 | 0.115 | 0.186 | 0.085 | 0.157 | 0.243 | −0.177 | −0.006 | 0.013 | −0.038 | 0.826 | 1.000 |
| | FR | 0.096 | −0.027 | −0.034 | −0.014 | 0.226 | −0.101 | 0.103 | 0.023 | 0.310 | 0.253 | 0.142 | 1.000 |

In the Shimen Reservoir, WT and pH were highly correlated with a correlation coefficient of 0.700 (Table 3).

In the Mingde Reservoir, the pairs of WT and conductivity, DO and pH, pH and conductivity, and rainfall and inflow were highly correlated with correlation coefficients of −0.644, 0.634, −0.649, and 0.826 respectively (Tables 3 and 4).

In the Fongshan Reservoir, the pairs of Chl-a and COD, DO and pH, SD and SS, TP and Conductivity, TP and AN, and conductivity and AN were highly correlated with correlation coefficients of 0.602, 0.700, −0.600, 0.854, 0.827, and 0.696, respectively (Tables 3 and 4).

The correlation coefficient of rainfall and inflow in the Mingde Reservoir as well as the correlation coefficient of TP with conductivity and TP with AN in the Fongshan Reservoir exceeded 0.8. However, when performing an MLR analysis, a high correlation coefficient between the independent variables causes the problem of collinearity in the regression results. This problem means that the lower the correlation between the independent variables, the more it reflects the relationship with dependent variables.

### 3.3. Analysis Result

Tables 5–7 show the regression analysis result after including time-lag and interaction terms from the Shimen, Mingde, and Fongshan Reservoir data, respectively. This study divides independent variables into "Basic" analysis without time lag, "Lag 1" data with one month lagged, and "Lag 2" data with two months lagged. For example, there is a time lag between an increase in water temperature, the growth of algae, the flow time of rainfall runoff to the reservoir, etc.

**Table 5.** Results from regression analyses of the Shimen Reservoir data.

|  | Coefficient | SE | *p*-Value |
| --- | --- | --- | --- |
| Intercept | 8.408 | 4.119 | 0.041 * |
| WT | 0.291 | 0.069 | 0.000 * |
| DO | 0.511 | 0.312 | 0.102 |
| SD | 0.028 | 0.125 | 0.822 |
| TP | −26.874 | 30.033 | 0.371 |
| pH | −1.078 | 0.199 | 0.000 * |
| Conductivity | −0.038 | 0.005 | 0.000 * |
| SS—Lag 1 | 0.033 | 0.013 | 0.014 * |
| COD | 0.478 | 0.123 | 0.000 * |
| AN | −54.684 | 22.944 | 0.017 * |
| Rainfall—Lag 1 | 0.047 | 0.020 | 0.015 * |
| Inflow | 0.001 | 0.001 | 0.075 |
| WT × COD | −0.025 | 0.005 | 0.000 * |
| TP × AN | 3817.198 | 788.920 | 0.000 * |
| S1 | 1.601 | 0.756 | 0.034 * |
| S2 | 0.656 | 0.524 | 0.211 |
| S3 | 1.605 | 0.614 | 0.009 * |
| $R^2$ | 0.517 | | |
| Obs. | 210 | | |

* $p < 0.05$.

According to the three conditions listed in Section 2.3.3, 'WT × COD' and 'TP × AN' were selected as interaction terms for the Shimen Reservoir regression model; 'WT × pH' and 'DO × TP' were the interaction terms for the Mingde Reservoir regression model. Since there was no significant interaction term for the Fongshan Reservoir, we used the result of the time-lag analysis as the final analysis result. The coefficient and correlation of WT, TP, COD, and AN in the Shimen and Mingde Reservoirs should be considered using interaction terms.

**Table 6.** Results from regression analyses of the Mingde Reservoir data.

|  | Coefficient | SE | *p*-Value |
|---|---|---|---|
| Intercept | −474.185 | 36.471 | 0.000 * |
| WT | 17.831 | 0.536 | 0.000 * |
| DO | −3.540 | 0.830 | 0.000 * |
| SD—Lag 2 | 1.692 | 0.721 | 0.019 * |
| TP | −1697.474 | 178.954 | 0.000 * |
| pH | 60.517 | 3.713 | 0.000 * |
| Conductivity—Lag 1 | 0.073 | 0.050 | 0.145 |
| SS—Lag 2 | 0.047 | 0.155 | 0.760 |
| COD | 1.502 | 0.127 | 0.000 * |
| AN | 76.633 | 16.141 | 0.000 * |
| Rainfall | −0.661 | 0.036 | 0.000 * |
| Inflow | 0.411 | 0.021 | 0.000 * |
| WT × pH | −2.256 | 0.079 | 0.000 * |
| DO × TP | 174.674 | 17.448 | 0.000 * |
| S1 | 2.862 | 2.468 | 0.246 |
| S2 | 23.801 | 0.672 | 0.000 * |
| S3 | 22.537 | 2.738 | 0.000 * |
| $R^2$ | 0.701 |  |  |
| Obs. | 102 |  |  |

* $p < 0.05$.

**Table 7.** Results from regression analyses of the Fongshan Reservoir data.

|  | Coefficient | SE | *p*-Value |
|---|---|---|---|
| Intercept | 417.270 | 62.137 | 0.000 * |
| WT | 5.237 | 1.654 | 0.002 * |
| DO—Lag 1 | −0.139 | 1.294 | 0.915 |
| SD | −40.372 | 12.974 | 0.002 * |
| TP—Lag 2 | 36.883 | 5.015 | 0.000 * |
| pH—Lag 1 | −55.004 | 11.780 | 0.000 * |
| Conductivity | −0.117 | 0.042 | 0.006 * |
| SS | −0.440 | 0.094 | 0.000 * |
| COD | 2.387 | 0.704 | 0.001 * |
| AN—Lag 1 | −6.933 | 4.499 | 0.123 |
| Rainfall | −0.959 | 0.982 | 0.329 |
| Inflow—Lag 1 | −0.324 | 0.317 | 0.307 |
| S1 | 6.626 | 13.378 | 0.620 |
| S2 | −32.329 | 7.933 | 0.000 * |
| S3 | −15.980 | 10.541 | 0.130 |
| $R^2$ | 0.605 |  |  |
| Obs. | 101 |  |  |

* $p < 0.05$.

For example, when using the interaction term 'TP × AN' for the Shimen Reservoir in Equation (6), the concentration of Chl-a is designated as the dependent variable $Y$, the concentration of TP is designated as the independent variable $X_1$, and the concentration of AN is designated as the independent variable $X_2$. $\beta_1$, $\beta_2$, and $\beta_3$ are the coefficients of TP, AN, and the 'TP × AN' interaction term, respectively.

Assuming the value of TP is 0.02 mg/L and AN is 0.03 mg/L, then 'TP × AN' is 0.0006 after multiplying the two. Inserting the above values and the coefficients '−26.874', '−54.684', and '3817.198' into Equation (6) results in a concentration of 0.112 (mg/L) Chl-a when other variables are fixed and the concentration of TP and AN are 0.02 and 0.03 mg/L (Equation (13)).

$$[(-26.874 \times 0.02)] + [(-54.684) \times 0.03] + (3817.198 \times 0.0006) = 0.112 \qquad (13)$$

When TP and AN both increase by 1 unit, the concentration of TP becomes 1.02 mg/L and AN becomes 1.03 mg/L, and, thus, the value of 'TP × AN' will become 1.0506. Then, the concentration of Chl-a is calculated as 3926.5 mg/L. This means that the unit value of Chl-a increases when other variables are fixed and TP and AN are both increased by 1 unit (Equation (14)).

$$(-26.874) + (-54.684) + 3817.198 + 3817.198 \times (0.02 + 0.03) = 3926.5 \qquad (14)$$

These results show that when evaluating the water quality trophic state, time-lag and the additive relationship of interactions between factors should also be taken into account to evaluate more accurately the potential for water eutrophication.

Note that the final model for each reservoir is different. This reflects the fact that the eutrophication process in reservoirs is a complex of many different factors, and the process is highly case-by-case. As shown in the introduction, influencing factors sometimes work in opposite directions in different cases from different studies. The result of this study further confirms that even reservoirs in Taiwan show very different patterns when it comes to the factors influencing the trophic state.

### 3.4. Standardization Coefficient

The regression coefficient of each factor was multiplied by the standard deviation of its data to form a 'standardized coefficient'. The basic amounts and units of each factor were different, and it is difficult to predict the dependent variable values based on only the regression coefficient value of each factor. The relative importance of each factor can be compared by standardizing the coefficients, making it is easier to intuitively understand the degree of influence of each independent variable on the dependent variable.

Tables 8–10 show the standardization coefficient of the Shimen, Mingde, and Fongshan Reservoirs. These results show the degree of influence of each factor on Chl-a while taking into account time-lag and variable interactions.

**Table 8.** Standardization coefficients of variables measured at the Shimen Reservoir.

|  | Coef. | SD | Std. Coef. ** | *p*-Value |
|---|---|---|---|---|
| WT | 0.291 | 4.413 | 1.284 | 0.000 * |
| DO | 0.511 | 1.085 | 0.554 | 0.102 |
| SD | 0.028 | 0.768 | 0.022 | 0.822 |
| TP | −26.874 | 0.011 | −0.296 | 0.371 |
| pH | −1.078 | 0.584 | −0.630 | 0.000 * |
| Conductivity | −0.038 | 26.936 | −1.024 | 0.000 * |
| SS—Lag 1 | 0.033 | 6.346 | 0.209 | 0.014 * |
| COD | 0.478 | 3.421 | 1.635 | 0.000 * |
| AN | −54.684 | 0.019 | −1.039 | 0.017 * |
| Rainfall—Lag 1 | 0.047 | 7.226 | 0.340 | 0.015 * |
| Inflow | 0.001 | 260.253 | 0.260 | 0.075 |

* $p < 0.05$. ** 'Std. Coef.': Is the factor regression coefficient multiplied by the standard deviation. All other factors are fixed, the degree of influence of each increase by one standard deviation of the factor affects the concentration of Chl-a.

At the Shimen Reservoir (Table 8), the influence of interactions must be considered when analyzing WT, TP, COD, and AN, so these variables will not be discussed separately. Other highly influential factors are conductivity (standardization coefficient = −1.024), secondly pH (standardization coefficient = −0.630), and lastly SS with a lag of one month (standardization coefficient = 0.209).

At the Mingde Reservoir (Table 9), the influence of interactions must be considered when analyzing WT, DO, TP, and pH, so these variables will not be discussed separately. Other factors with a high influence were inflow (standardization coefficient = 11.295), followed by rainfall (standardization coefficient = −11.166), and lastly SD with two months lag (standardization coefficient = 0.548).

**Table 9.** Standardization coefficients of variables measured at the Mingde Reservoir.

|  | Coef. | Std. Dev. | Std. Coef. ** | *p*-Value |
|---|---|---|---|---|
| WT | 17.831 | 5.012 | 89.369 | 0.000 * |
| DO | −3.540 | 2.16 | −7.646 | 0.000 * |
| SD–Lag 2 | 1.692 | 0.324 | 0.548 | 0.019 * |
| TP | −1697.474 | 0.006 | −10.185 | 0.000 * |
| pH | 60.517 | 0.499 | 30.198 | 0.000 * |
| Conductivity—Lag 1 | 0.073 | 37.782 | 2.758 | 0.145 |
| SS—Lag 1 | 0.047 | 3.883 | 0.183 | 0.760 |
| COD | 1.502 | 2.967 | 4.456 | 0.000 * |
| AN | 76.633 | 0.029 | 2.222 | 0.000 * |
| Rainfall—Lag 1 | −0.661 | 16.893 | −11.166 | 0.000 * |
| Inflow | 0.411 | 27.482 | 11.295 | 0.000 * |

* $p < 0.05$. ** 'Std. Coef.': Is the factor regression coefficient multiplied by the standard deviation. All other factors are fixed, the degree of influence of each increase by one standard deviation of the factor affects the concentration of Chl-a.

**Table 10.** Standardization coefficients of variables measured at the Fongshan Reservoir.

|  | Coef. | Std. Dev. | Std. Coef. ** | *p*-Value |
|---|---|---|---|---|
| WT | 5.237 | 3.093 | 16.198 | 0.002 * |
| DO—Lag 2 | −0.139 | 2.699 | −0.375 | 0.915 |
| SD | −40.372 | 0.218 | −8.801 | 0.002 * |
| TP—Lag 2 | 36.883 | 0.440 | 16.229 | 0.000 * |
| pH—Lag 1 | −55.004 | 0.324 | −17.821 | 0.000 * |
| Conductivity | −0.117 | 177.133 | −20.725 | 0.006 * |
| SS | −0.440 | 8.298 | −3.651 | 0.000 * |
| COD | 2.387 | 8.901 | 21.247 | 0.001 * |
| AN—Lag 1 | −6.933 | 1.493 | −10.351 | 0.123 |
| Rainfall | −0.959 | 6.328 | −6.069 | 0.329 |
| Inflow—Lag 1 | −0.324 | 6.648 | −2.154 | 0.307 |

* $p < 0.05$. ** 'Std. Coef.': Is the factor regression coefficient multiplied by the standard deviation. All other factors are fixed, the degree of influence of each increase by one standard deviation of the factor affects the concentration of Chl-a.

Data from the Fongshan Reservoir (Table 10) showed that COD had the greatest influence on Chl-a (standardization coefficient = 21.247), which was followed by conductivity (standardization coefficient = −20.725) and lastly SS (standardization coefficient = −3.651).

To summarize, at the Shimen Reservoir, Chl-a was significantly and immediately affected by WT, pH, Conductivity, COD, and AN, and significantly, but not immediately, affected by SS and rainfall. DO, SD, TP, and inflow did not significantly affect Chl-a at the Shimen Reservoir. Chl-a at the Mingde Reservoir was significantly and immediately affected by WT, DO, TP, pH, COD, AN, rainfall, and inflow, while the effect of SD was significant but not immediate. Conductivity and SS did not significantly affect Chl-a at the Mingde Reservoir. WT, SD, Conductivity, SS, and COD significantly and immediately affected Chl-a at the Fongshan Reservoir; TP and pH also significantly affected Chl-a, but not immediately; while the effects of DO, AN, rainfall, and inflow were not significant.

### 3.5. Correlation of Factors

Table 11 illustrates the correlation results of each weather and water quality factor. Table 11 clearly indicates that the correlation of WT in the Shimen and Mingde Reservoirs needs to be considered within an interaction and is significantly positive in the Fongshan Reservoir. The correlation of DO in the Mingde Reservoir needs to be considered within an interaction, and it is not significantly correlated in the Shimen and Fongshan Reservoirs. The correlation of SD is significantly positive in the Mingde Reservoir and significantly negative in the Fongshan Reservoir, but it is not significantly correlated in the Shimen Reservoir. The correlation of TP in the Shimen and Mingde Reservoirs needs to be considered within an interaction and is significantly positive in the Fongshan Reservoir. The

correlation of pH in the Mingde Reservoir needs to be considered within an interaction and is significantly negative in both the Shimen and Fongshan Reservoirs. The correlations of conductivity in the Shimen and Fongshan Reservoirs are significantly negative but not significant in the Mingde Reservoir. The correlation of SS is significantly positive in the Shimen Reservoir and significantly negative in the Fongshan Reservoir, but there is no significant correlation in the Mingde Reservoir. The correlation of COD in the Shimen Reservoir needs to be considered within an interaction, but it is significantly positive in both the Mingde and Fongshan Reservoirs. The correlation of AN in the Shimen Reservoir needs to be considered within an interaction, and it is significantly positive in the Mingde Reservoir but not significantly correlated in the Fongshan Reservoir. The correlation of rainfall is significantly positive in the Shimen Reservoir and significantly negative in the Mingde Reservoir, but there is no significant correlation in the Fongshan Reservoir. The correlation of inflow is significantly positive in the Mingde Reservoir, but there is no significant correlation in the Shimen and Fongshan Reservoirs.

**Table 11.** Correlation of factors from the Shimen, Mingde, and Fongshan Reservoirs.

|  | Shimen | Mingde | Fongshan |
|---|---|---|---|
| WT | I | I | + |
| DO | X | I | X |
| SD | X | + | − |
| TP | I | I | + |
| pH | − | I | − |
| Conductivity | − | X | − |
| SS | + | X | − |
| COD | I | + | + |
| AN | I | + | X |
| Rainfall | + | − | X |
| Inflow | X | + | X |
| WT × COD | − | N | N |
| TP × AN | + | N | N |
| WT × pH | N | − | N |
| DO × TP | N | + | N |

⌈ I ⌋: The correlation of factors needs to take into account interaction relationships. ⌈ X ⌋: Factors were not significantly correlated. ⌈ + ⌋: Factors have a significant positive correlation. ⌈ − ⌋: Factors have a significant negative correlation. ⌈ N ⌋: The variable is not relevant for the reservoir.

There was no deterministic causality between climate and water quality variables. For example, the pH in the Fongshan Reservoir is negatively correlated with Chl-a, but Zang (2011) shows that Chl-a is positively correlated with pH and DO [29]. The same case as Blumberg (1990) finds that WT is negatively correlated with DO [30], but Chen (2007) shows that WT is positively correlated with DO [31]. In another, case Watson (2016) shows that Chl-a is negatively correlated with DO [32], but Zang (2011) shows a positive correlation [29].

The interaction combinations for the Shimen Reservoir are 'WT × COD' and 'TP × AN', and for the Mingde Reservoir, they are 'WT × pH' and 'DO × TP'. There were no significant correlation interactions for the Fongshan Reservoir. Note that temperature and total phosphorus are the only two factors that have a positive influence among all three reservoirs. However, the effect of temperature negatively interacts with COD in Shimen and with pH in Mingde, making the effect of temperature on the trophic state actually more minor than expected. On the other hand, the interaction terms related to the total phosphorus in Shimen and Mingde are magnifying the effect. This result further supports that total phosphorus is the main factor for the trophic state.

To summarize, these results indicate that the influencing factors of the trophic state in reservoirs defer from case to case; thus, it is difficult to find a one-size-fits-all equation to be perfectly suitable in all cases.

## 4. Conclusions

The main factor influencing the three reservoirs is total phosphorus. At the Shimen and Mingde Reservoirs, in particular, the interactive effect of TP with other factors on the water quality trophic state was greater than that of TP alone, indicating that more attention should be paid to the interaction effect between the influencing factors. However, there is no significant interaction effect found to further aggravate the trophic state between weather and water quality factors. In the case of these three reservoirs in Taiwan, an additional deterioration of eutrophication from the climate-change-related interaction effect is not a concern.

The analysis of characteristics influenced by time lags and the analysis of the interactions between factors provide a deeper understanding of the correlation between each factor and the degree to which they influence the water quality trophic state. Furthermore, the length of the time lag and the significant combinations of influencing factors vary from reservoir to reservoir, indicating that the patterns of eutrophication might differ according to different reservoir conditions. These results imply that factors influencing the tropic state in a reservoir might vary by reservoir type, geological and meteorological conditions, as well as other potential factors. In other words, forming a model that describes the tropic state for a reservoir is highly case sensitive. The perfect solution of a one-size-fits-all model might not exist. Researchers should carefully review all possible factors before finalizing a model.

In this study, the $R^2$ values of the MLR model developed for the three reservoirs were all above 0.5, indicating that the regression model for each reservoir explains more than half of the cause of the water quality trophic state. The results indicate that the regression model developed during this study and the methods used are both feasible for assessing the water quality trophic state.

**Author Contributions:** Conceptualization, L.-H.C.H.; methodology, C.-W.H. and L.-H.C.H.; software, C.-W.H.; validation, L.-H.C.H.; writing—original draft preparation, C.-W.H.; writing—review and editing, L.-H.C.H.; visualization, C.-W.H. All authors have read and agreed to the published version of the manuscript.

**Funding:** Ministry of Science and Technology, Taiwan: 109-2221-E-033-004-MY2.

**Data Availability Statement:** Environmental Protection Administration, Environmental Water Quality Information (https://wq.epa.gov.tw/EWQP/en/Default.aspx, accessed on 31 August 2021); Water Resources Agency, Disaster Prevention Information Service (https://fhy.wra.gov.tw/fhy/Monitor/Reservoir, accessed on 31 August 2021).

**Acknowledgments:** The authors would like to thank Uni-edit (www.uni-edit.net, 31 August 2021) for editing and proofreading this manuscript.

**Conflicts of Interest:** The authors declare no conflict of interest.

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
