# Peer review of "Analysis of Factors Influencing the Trophic State of Drinking Water Reservoirs in Taiwan"

_water, doi:10.3390/w13223228_

Round 1

Reviewer 1 Report

The work may be of interest to some part of readership, although no new findings were derived. The fact that phosphorus is a driver to eutrophication has been well known and the fact that it is difficult to predict interactions between various variables  in big water systems (or to indicate cause and effect relationship) – too. However, the work may be of local interest and also from the point of view of the use of MLR analysis.

I suggest major changes because in its present form the manuscript does not contain Discussion and no literature citations were included. The authors just present their results and this is not sufficient to demonstrate a scientific importance of their work (for larger readership).

In Abstract

“the influence of total phosphorus, when interacting with other factors, on water quality nutrition is more serious than that of total phosphorus per se, which implies the influence of climate change on the eutrophic condition could be underestimated.

Further, there was no deterministic causality between climate and water 16 quality variables.”

  1. Not clear what is ‘’water quality nutrition’’. Maybe “water enrichment with nutrients”?
  2. The sentence needs re-phrasing because in the first part the authors speak about many other factors, a part from total phosphorus, but in the second part it is unclear why they mention only climate change from among other factors.
  3. They results showed that climate change is not a main factor for these three reservoirs (Lines 372-373) and in Abstract they say that the ‘’influence of climate change on the eutrophic conditionmay be underestimated.’’

The overall text of the abstract needs revision, as the authors first indicate the importance of climate variables (“which implies the influence of climate change on the eutrophic condition could be underestimated.”) but then say that “there was no deterministic causality between climate and water quality variables” and then “The factors that were not significant are unlikely to have zero influence”.

Lines 19-21

‘’The background environment, surrounding facilities, and the degree of development in the 19 water catchment area of each reservoir are different, resulting in different factors influencing the 20 reservoirs.’’

Nothing is discussed about these factors in Discussion and, so why they are mentioned in the Abstract? This sentence can be a part of Discussion (which is practically missing).

For me the Abstract is not logical, is not convincing and is rather vague, without reflecting the actual work done. In my opinion, it needs to be re-written.

Also, do the authors speak about total phosphate (organic and inorganic phosphates) or total phosphorus (phosphates and phosphorus contained in suspended matter)? They say in Abstract and Discussion ‘’total phosphorus’’, but as independent variables they say ‘’total phosphate’’.

Lines 35-38

Since the 1940s, a substantial population increase, land-use intensification, and the use of agricultural fertilizers from developed countries [11], as well as the use of detergents containing phosphate compounds since the 1950s, have accelerated eutrophication of waterbodies worldwide [12].

I suggest deleting “worldwide” as the sentence in its present form, says that population increase, etc. in developed countries have resulted in the eutrophication worldwide. One may ask does the input of untreated sewage, etc. from under-depeloped countries did not contribute to the eutrophication locally and globally?

Lines 39-43

The consequence of algal blooms listed here should be properly referenced, unless the authors speak about their own results.

Lines 111-112.

Aim of work must be stated in Introduction, but not on Material and methods. Thus, the aim must be removed from here.

Lines 118-141.

The authors never mentioned what program was used to run MLR analysis. Please also indicate the version of the program.

The authors also did not mention if they checked for the MLR assumptions (normality and homoscedacity of regression residuals). Also, Pearson correlation analysis that is used in MLR requires a linear relationship between the independent and dependent variables. Were these checks performed? This at least should be mentioned in the text that assumption of MLR were verified.

Lines 132-135 must go into Discussion section and discussed with regard to the obtained results. Here the authors can briefly mention why they chose lag variables.

Line 141

Not clear what is meant by “not discussed’’ in the sentence ”Basic analysis data were selected as a representative term, but not discussed”.

Lines 143-144

“We also test whether the influence weather and water quality factors have on water quality is affected by their interaction.”

Not clear. Please correct the grammar.

Section 4.

Discussion is practically missing and no references to the works of others supporting the present results are included. Some parts in the Results sections can be moved to Discussion, e.g., 324-332, but I do recommend expanding Discussion and including comparisons with other works.

CTSI is very briefly mentioned. Instead, the authors should clearly show why this index cannot be applied to these reservoirs.

The sentence “Weighing one-third in the CTSI index is clearly insufficient evidence to suggest that a reservoir is in a tropic state” is unclear.

If journal allows, Results and Discussion can be combined. In this case, Section 4 can be as Conclusions, but in a shortened form.

Cited works are missing!

Reviewer 2 Report

The paper of Cheng-Wei Hung and Lin-Han Chiang Hsieh “Analysis of Factors Influencing the Trophic State of drinking water Reservoirs in Taiwan” is interesting and actual investigations devoted to the application of the multiple linear regression model for assessing the quality of water reservoirs depended on different factors. Chlorophyll a was used as an indicator to illustrate the degree of eutrophication, and data were analyzed using multiple linear regressions (MLR) including time lags and variable interactions. Three reservoirs (Shimen, Mingde, and Fongshan Reservoirs in Taiwan) have been considered by the authors. Eutrophication is an environmental pollution problem that occurs in natural water bodies. This is an important ecological investigation, but there are some comments and remarks:

  1. The authors noted that the main factor influencing the eutrophication of water basins is phosphorus pollution. But it is unclear about anthropogenic factors which are phosphorus sources. Are there any differences between water reservoirs? It is unclear the meсhanism of phosphorus income for each of the water reservoirs (Shimen, Mingde, and Fongshan). Which additional anthropogenic factors influenced the pollution?  
  2. If phosphorus comes in as a result of the outflow of fertilizers from fields how much is for each of the reservoirs. Is it possible to control its income and decrease the eutrophication of the water basin? Which anthropogenic and technogenic objects are located near reservoirs and what is the influence on the pollutions of basins? 
  1. It is a necessity to present the map of the location of reservoirs under consideration. It is important for the understanding of local hydrographical features basins and processes of eutrophication and the problems connected with it.
  2. It is unclear the events of negative correlations between factors. Authors should more detail to describe how to influence different factors on decreasing the level of eutrophication.    
  3. It would be appropriate to describe the influence of temperature factors, especially in Summer periods when the high temperature prevails. The long-time predictions would be important also.
  4. Assumed on the data obtained on the modeling of MLR of the eutrophication of water reservoirs, authors should in detail consider how could be decreased factors and the eutrophic level in the different water reservoirs. It is necessary to give some recommendations for each reservoir on how does the level of eutrophication could be decreased. 

Round 2

Reviewer 1 Report

n/a

Reviewer 2 Report

The authors took into account the comments. The article was improved and can be accepted after minor revision (corrections to minor methodological errors and text editing).